# Synthesis, Characterization, and Evaluation of Nanoparticles Loading Adriamycin Based on 2-Hydroxypropyltrimethyl Ammonium Chloride Chitosan Grafting Folic Acid

**DOI:** 10.3390/polym13142229

**Published:** 2021-07-06

**Authors:** Yingqi Mi, Jingjing Zhang, Lulin Zhang, Qing Li, Yuanzheng Cheng, Zhanyong Guo

**Affiliations:** 1Key Laboratory of Coastal Biology and Bioresource Utilization, Yantai Institute of Coastal Zone Research, Chinese Academy of Sciences, Yantai 264003, China; yqmi@yic.ac.cn (Y.M.); jingjingzhang@yic.ac.cn (J.Z.); qli@yic.ac.cn (Q.L.); 2Center for Ocean Mega-Science, Chinese Academy of Sciences, 7 Nanhai Road, Qingdao 266071, China; 3University of Chinese Academy of Sciences, Beijing 100049, China; 4School of Pharmacy, Weifang Medical University, Weifang 261053, China; zhang554028@163.com

**Keywords:** chitosan, nanoparticles, drug release, cytotoxicity

## Abstract

Chitosan nanoparticles have been considered as potential candidates for drug loading/release in drug delivery systems. In this paper, nanoparticles (HACAFNP) loading adriamycin based on 2-hydroxypropyltrimethyl ammonium chloride chitosan grafting folic acid (HACF) were synthesized. The surface morphology of the novel nanoparticles was spherical or oval, and the nanoparticles exhibited a relatively small hydrodynamic diameter (85.6 ± 2.04 nm) and positive zeta potential (+21.06 ± 0.96 mV). The drug release of nanoparticles was assayed and represented a burst effect followed by a long-term steady release. Afterward, the antioxidant efficiencies of nanoparticles were assayed. In particular, the target nanoparticles exhibited significant enhancement in radical scavenging activities. Cytotoxicities against cancer cells (MCF-7, BGC-823, and HEPG-2) were estimated in vitro, and results showed nanoparticles inhibited the growth of cancer cells. It’s worth noting that the inhibition index of HACAFNP against BGC-823 cells was 71.19% with the sample concentration of 25 μg/mL, which was much higher than the inhibitory effect of ADM. It was demonstrated that the novel nanoparticles with dramatically enhanced biological activity, reduced cytotoxicity, and steady release could be used as the practical candidates for drug loading/release in a delivery system.

## 1. Introduction

Despite the rapid development of technology and medicine in the 21st century, cancer is still one of the major diseases that can threaten the safety and health of human life all over the world [1,2]. There are three kinds of traditional cancer treatments: surgical operation, chemotherapy, and radiotherapy [3]. Depending on the type of cancer, the stage of development, and the general condition of patient, the treatment strategy may vary. Among them, chemotherapy plays an important role in cancer therapy. Although conventional chemotherapy has achieved success to some extent, it is inevitably limited by poor utilization rate, development of multiple drug resistances, lack of targeting, toxic and adverse effects, and so on [4]. In addition, in traditional drug delivery systems, chemotherapeutic drugs present a “burst effect”, resulting in short drug active times, low therapeutic indices, and increased liver nephridium burden [5]. Taking adriamycin (ADM) as an example, it is an anthracycline antitumor antibiotic fermented by *streptomyces peucetius var. caesius*, which is one of the most efficient broad-spectrum antineoplastic drugs for a great variety of cancers [2]. It works by impeding the replication of DNA and causing DNA damage to achieve anti-tumor effects [6]. However, the clinical application of ADM has been restricted by a dose-limiting application, which may cause life-threatening cardiomyopathy and heart failure [7,8].

In recent years, nanoparticles have attracted sufficient attention in the research of novel drug formulations. It has been reported that the nano encapsulation of chemotherapeutic drugs is a good and practical method in order to enhance the physical stability and bioactivity [9]. On the one hand, nanoparticles can be absorbed by cells via penetrating tissue gaps, passing through the body’s smallest capillaries. On the other hand, nanoparticles can protect the loading drugs from destruction via environmental factors. Furthermore, nanoparticles have emerged as promising drug carriers due to their unique advantages such as controlled release in order to prolong bioavailability and half-life, decreased toxic and adverse effects, improved bioavailability and efficacy, and enhanced water solubility [5,10]. Therefore, the development of chemotherapeutic drug delivery methods such as using nanoparticles has been proposed as an ideal approach in order to increase the efficacy of chemotherapy and decrease toxic and adverse effects. There has been increasing interest in seeking drug carriers with good biocompatibility, biodegradability, bioactivity (anti-inflammatory, antioxidant activity), and no immune protoplasm, which can control drug release, prolong drug efficacy, and improve drug utilization.

Chitosan, a polysaccharide of cationic β-(1→4)-2-amino-2-deoxy-*D*-glucan, is prepared commercially by partial chemical deacetylation of chitin in the alkaline reaction system [11,12,13,14]. As a kind of marine active substance, chitosan has drawn attention in biomedical and biotechnological fields since it possesses excellent biological properties, including biocompatibility, non-toxicity, biodegradability, antioxidant activity, antimicrobial activity, and so on [15,16]. Therefore, it has been reported that chitosan nanoparticles can be served as practical drug carriers with the advantage of a controlled and slow drug release, which can reduce toxic and adverse effects and enhance the efficacy of chemotherapy [9,17]. Iurciuc-Tincu et al. developed novel, high drug-loaded complex microparticles through ionic cross-linking and polyelectrolyte complexation. The results showed that the microparticles, administered orally, can be used to target the colon for controlled release of curcumin [18]. Rata et al. investigated a kind of nanocapsule based on carboxymethyl chitosan bearing AS 1411 aptamer and poly (N-vinylpyrrolidone-alt-itaconic anhydride) loading 5-fluorouracil. The nanocapsules loading 5-fluorouracil were proved to be able to induce cell senescence and apoptosis MCF-7 cells [19]. Guilherme and Breno reported on a kind of chitosan nanoparticle loading oxaliplatin, which was used as a mucoadhesive topical treatment of oral tumors [20]. They found that these kinds of chitosan nanoparticles loading oxaliplatin cause a drug burst effect at first, followed by a longer-term controlled release. The chitosan nanoparticles formulation enhanced threefold the drug utilization. Meanwhile, it is critical for polysaccharides as nanomaterials with good biological activity to avoid infection and immunoreaction [21,22]. Since 2-hydroxypropyltrimethyl ammonium chloride chitosan (HACC) is more soluble and active than chitosan but has the same physical and chemical properties, it may be used as a drug carrier with good antioxidant activity to synthesize an efficient controlled drug release system [17]. In addition, folic acid, also known as vitamin B9, has reported to be a water-soluble vitamin with high antioxidant activity and low immunogenicity [23,24]. It is highly expressed in cancer cells due to its highly specific affinity with folate receptors [25]. Therefore, we hypothesize that nanoparticles loading ADM based on HACC grafting folic acid could be a practical drug carrier with the advantages of slow and controlled drug release, enhanced antioxidant activity, antitumor activity, and reduced toxicity.

In this present work, the objective was to develop novel nanoparticles loading ADM based on HACC grafting folic acid through ionic gelation with the aid of sodium tripolyphosphate (TPP), and to evaluate the morphology, properties, and activities of nanoparticles. The obtained nanoparticles were confirmed by means of Fourier-transform infrared spectroscopy (FTIR), dynamic light scattering (DLS), zeta potential measurements, and scanning electron microscopy (SEM). In addition, the degrees of substitution (DS) of chitosan derivatives were calculated via ^1^H nuclear magnetic resonance (NMR) spectroscopy. The entrapped efficiency and drug loading of ADM in nanoparticles were quantified by measuring the amount of ADM remaining in the supernatant after centrifugation. Meanwhile, the drug release of the novel nanoparticles was assayed in vitro. Afterward, the antioxidant efficiencies (superoxide-radical scavenging activity and DPPH radical scavenging activity) of the novel nanoparticles were assayed. The cytotoxicities against three kinds of cancer cells (MCF-7, BGC-823, and HEPG-2) and normal cells (HRT-8) were estimated in vitro.

## 2. Materials and Methods

### 2.1. Materials

Chitosan with an average molecular weight (MW) of 200 kDa and degree of deacetylation of 85% was obtained from Qingdao Baicheng Biochemical Corp. (Qingdao, China). The materials, including 3-chloro-2-hydroxypropyltrimethyl ammonium chloride, sodium tripolyphosphate (TPP), folic acid, adriamycin hydrochloride, and 2,2-diphenyl-1-picrylhydrazyl (DPPH) were purchased from Sinopharm Chemical Reagent Co., Ltd. (Shanghai, China). Isopropyl alcohol, sodium hydroxide, and ethanol were provided by Sigma-Aldrich Chemical Corp. (Shanghai, China). All the chemical reagents and solvents were obtained from commercial sources and used as received.

### 2.2. Preparation of Nanoparticles

The nanoparticles loading ADM based on HACC grafting folic acid through ionic gelation with the aid of sodium tripolyphosphate were synthesized via multistep reactions (Scheme 1).

#### 2.2.1. Preparation of Chitosan Derivatives

HACC was synthesized as previously report [11,26]. Firstly, chitosan (1.61 g, 10 mmol) was dissolved in 60 mL of isopropanol ahead of adding 4 mL of 40% NaOH (*w/v*) aqueous solution, and the reaction mixture was stirred vigorously at 60 °C at a speed of 3500 rpm. After 4 h, 12.5 mL of 60% (*w/v*) 3-chloro-2-hydroxypropyl trimethyl ammonium chloride solution was added to the previous reaction system with stirring at a speed of 3500 rpm at 75 °C for 10 h. The right amount of hydrochloric acid was then added in order to adjust the pH of mixture to 7. Afterward, plenty of anhydrous ethanol was used to precipitate. The precipitate was purified by rinsing repeatedly with 80% (*w/v*) ethanol, and HACC was obtained by lyophilizing in vacuo. Finally, HACC (0.935 g, 3 mmol) was dissolved in 30 mL of a deionized water solution of sodium folate for the purpose of replacing the chloride ions with folate sodium anions with the aid of ion exchange. After 6 h, a clear solution was obtained and dialyzed against deionized water with a dialysis tube (molecular weight cutoff, MW CO: 500 Da) in order to eliminate the excess folate, Na^+^, and Cl^−^. After 48 h of dialyzing, the residual solution was dried in vacuo to obtain HACC grafting folic acid (HACF).

#### 2.2.2. Preparation of Nanoparticles via HACC and HACF

##### Preparation of Nanoparticles via HACC

Nanoparticles (HACANP) loading ADM based on HACC were synthesized via ionic gelation with the aid of sodium tripolyphosphate. Through experiments, the optimal preparation conditions were selected based on hydrodynamic diameter, size potential, encapsulation efficiency, and drug loading. In a nutshell, HACANP was prepared by the following method. HACC (0.2 g, 0.5 mmol) was dissolved in 100 mL of deionized water at 25 °C overnight. After sterilizing via a 0.22 μm filter, 5 mL of a diluted adriamycin hydrochloride solution was added drop by drop. Subsequently, 20 mL of a TPP solution (2 mg/mL) was dropped into the front mixture drop by drop with stirring at a speed of 2000 rpm at 25 °C. The nanoparticles were separated by centrifugation at 12,000 rpm for 10 min and lyophilized for further characterization, release, and bioactivity study. Meanwhile, the blank nanoparticles (HACNP) without loading ADM were prepared according to the above method by using deionized water instead of a diluted adriamycin hydrochloride solution.

##### Preparation of Nanoparticles via HACF

Nanoparticles (HACAFNP) loading ADM based on HACC grafting folic acid were also synthesized via ionic gelation with the aid of sodium tripolyphosphate. Through experiments, the optimal preparation conditions were selected based on hydrodynamic diameter, size potential, encapsulation efficiency, and drug loading. For HACAFNP, 100 mL of HACF (2 mg/mL) was stirred overnight at 25 °C at a speed of 2000 rpm. After sterilizing via a 0.22 μm filter, 5 mL of a diluted adriamycin hydrochloride solution and 20 mL of a TPP solution (2 mg/mL) were added drop by drop. The nanoparticles were separated via centrifugation at 12,000 rpmn for 10 min and lyophilized for further characterization, release, and bioactivity study. Meanwhile, the blank nanoparticles (HACFNP) without loading ADM were prepared according to the above method by using deionized water instead of a diluted adriamycin hydrochloride solution.

### 2.3. Characterization of Chitosan Derivatives and Nanoparticles

#### 2.3.1. Fourier-Transform Infrared Spectroscopy

The infrared spectra of chitosan, chitosan derivatives, and nanoparticles were obtained using a Nicolet iS 50 Fourier-Transform Infrared Spectrometer (Thermo Fisher Scientific, Waltham, MA USA) with transmission mode covering the frequency ranging from 4000 to 400 cm^−1^ at 25 °C. All the tested samples were scanned 32 times with a resolution of 4.0 cm^−1^ using the KBr pellet method (the weight ratio 1:100).

#### 2.3.2. H Nuclear Magnetic Resonance Spectroscopy of Chitosan Derivatives

The ^1^H NMR spectra of chitosan and chitosan derivatives were performed by a Bruker AVIII-500 Spectrometer (500 MHz, purchased from Bruker Tech. and Serv. Co., Ltd., Beijing, China) at 25 °C. All samples were dissolved in 0.6 mL D_2_O with a concentration of 20 mg mL^−1^ for measurement. The degrees of substitution (DS) of chitosan derivatives were calculated via ^1^H NMR spectroscopy. For example, the DS of HACF was calculated using the following equation:(1)$$DS\left(\%\right)=\frac{\frac{I_{H_{ij,}\;\mathrm{HACF}}}{2}}{\frac{I_{H_{d,}\mathrm{HACF}}}{9}}\times 100$$
where $I_{H_{ij,}\;\mathrm{HACF}}$ is the integral values of the *H**_ij_* on benzene ring; $I_{H_{d,}\mathrm{HACF}}\;$ is the integral values of the hydrogen atom bonded to N^+^(CH_3_)_3_ group; 2 means the number of protons in *H**_ij_* on HACF, and 9 means the number of protons in *H**_d_* on HACF.

#### 2.3.3. Hydrodynamic Diameter (nm), Zeta Potential (mV), and Nanoparticle Stability

The hydrodynamic diameter and zeta potential of nanoparticles were characterized using a nanometer particle size measuring instrument (Litesizer 500, Anton Paar Instruments, Graz, Austria), dynamic light scattering, and electrophoretic light scattering (ELS). All samples were measured three times for average value at 25 °C. Meanwhile, nanoparticle stability was also measured by the nanometer particle size measuring instrument.

#### 2.3.4. Morphology

The morphology of nanoparticles was characterized via scanning electron microscopy (SEM, S-4800, Hitachi, Tokyo, Japan). To be brief, the dried nanoparticles were put on a sample-holder and coated with gold for observation. Images were acquired by a scanning electron microscope operating at 3.0 kV.

### 2.4. Entrapped Efficiency and Drug Loading of Nanoparticles

The encapsulation efficiency and drug loading of freeze-dried nanoparticles (HACAFNP) loading ADM based on HACF were quantified by measuring the amount of free ADM remaining in the supernatant after centrifugation at 12,000 rpm for 10 min [27]. The ADM remaining in the supernatant was determined via an ultraviolet spectrophotometer (UV spectrophotometer, T6, Pgeneral, Beijing, China) with a measuring absorption at 480 nm according to the concentration (C)–absorbance (A) equation (A = 0.00387 C + 0.01321, R^2^ = 0.999). Three replicates for nanoparticles were measured, and the encapsulation efficiency and drug loading efficiency were calculated as follows:(2)$$\mathrm{Encapsulation}\;\mathrm{efficiency}\;(\%)=\frac{m_{\;\mathrm{total}\;\mathrm{ADM}}-m_{\;\mathrm{free}\;\mathrm{ADM}}}{m_{\;\mathrm{total}\;\mathrm{ADM}}}\times 100$$
(3)$$\mathrm{Drug}\;\mathrm{loading}\;(\%)=\frac{m_{\;\mathrm{total}\;\mathrm{ADM}}-m_{\;\mathrm{free}\;\mathrm{ADM}}}{m_{\;\mathrm{total}\;\mathrm{NP}}}\times 100$$
where $m_{\;\mathrm{total}\;\mathrm{ADM}}$ is the total weight of ADM used for the preparation of nanoparticles, $m_{\;\mathrm{free}\;\mathrm{ADM}}$ is the weight of free ADM remaining in the supernatant after centrifugation, and $m_{\;\mathrm{total}\;\mathrm{NP}}$ is the weight of nanoparticles.

### 2.5. Swelling Degree

The swelling degree (Q%) of chitosan nanoparticles was assessed following Anca’s methods [19]. Briefly, dry samples (*W*_0_, 0.05 g) were immersed in 5 mL of a swelling agent at 37 °C. According to predetermined intervals, the swelling agent was removed through filtration, and the chitosan nanoparticle surfaces were buffered via filter papers in order to remove the excess of swelling agent. The weight of the swollen samples (*W*_1_) was measured using analytical balance. After weighing, 5 mL of the swelling agent was reintroduced. Three replicates for nanoparticles were measured, and the swelling degree was calculated as follows:(4)$$\mathrm{Swelling}\;\mathrm{degree},\;Q(\%)=\frac{W_{\;1}-W_{\;0}}{W_{\;0}}\times 100$$

### 2.6. Evaluation of Sustained Release Performance

The sustained release performance of nanoparticles was measured using a RC1207DP dissolution tester (Tianjin, China) and ultraviolet spectrophotometer (UV spectrophotometer, T6, Pgeneral, Beijing, China) [28,29]. Firstly, 800 mL of phosphate buffer (pH = 7.4) was utilized as the dissolution media to ensure the internal environment. The 50 mg of freeze-dried nanoparticles was then dissolved in deionized water using a dialysis tube with a speed of 100 rpm at 37 °C. Finally, according to predetermined intervals, 2 mL of dissolution media was removed for released ADM determination using an ultraviolet spectrophotometer at 480 nm according to the concentration (C)–absorbance (A) equation (A = 0.00387 C + 0.01321, R^2^ = 0.999), and the same volume of phosphate buffer was compensated immediately. The release percentage of ADM was calculated by dividing the cumulative amount of released ADM at each sampling time point (*M**_t_*) to the original weight of ADM (*M*_0_). The release percentage was calculated using the following equation:(5)$$\mathrm{Release}\;\mathrm{percentage}\;(\%)=\sum_{t=0}^{t}\frac{M_{t}}{M_{0}}\times 100$$

### 2.7. Antioxidant Assays

#### 2.7.1. Superoxide-Radical Scavenging Activity Assay

The superoxide-radical scavenging ability of the chitosan nanoparticles was assessed following Tan’s methods [16]. Firstly, the tested samples were dissolved in deionized water to ensure the initial concentration of 10 mg/mL. Different volumes of initial sample solution (30, 60, 120, 240, and 480 μL) were then transferred to a centrifuge tube and deionized water was added to ensure a volume of 1.5 mL. Afterwrad, the 3 mL of total reaction mixture, involving 1.5 mL of tested samples, 0.5 mL of nitro blue tetrazolium (NBT, 72 μM), 0.5 mL of phenazine methosulfate (PMS, 30 μM), and 0.5 mL of nicotinamide adenine dinucleotide reduced (NADH, 338 μM) in Tris-HCl buffer (16 mM, pH 8.0), was incubated at 25 °C for 5 min. The absorbance was measured at 560 nm, and three replicates for every tested sample, control, and blank were recorded. The superoxide-radical scavenging effect was calculated using the following equation:(6)$$\mathrm{Scavenging}\;\mathrm{effect}\;\left(\%\right)=\left[1-\frac{A_{\mathrm{sample}\;560\;\mathrm{nm}}-A_{\mathrm{control}\;560\;\mathrm{nm}}}{A_{\mathrm{blank}\;560\;\mathrm{nm}}}\right]\times 100$$
where A_sample 560 nm_ is the absorbance of the samples, A_control 560 nm_ is the absorbance of the control (NADH was substituted with distilled water), and A_blank 560 nm_ is the absorbance of the blank (samples were substituted with distilled water).

#### 2.7.2. DPPH Radical Scavenging Ability Assay

The DPPH radical scavenging ability of the chitosan nanoparticles was tested according to the following method [30]. Two milliliters of a DPPH ethanol solution and tested samples with initial concentrations of 10 mg/mL (0.03, 0.06, 0.12, 0.24, 0.48 mL) were incubated at 25 °C for half an hour. Moreover, the tested samples were replaced by deionized water in the blank group, and ethanol was used instead of the 2 mL DPPH solution in the control group. Later, the absorbance of the DPPH radical was recorded at 517 nm. Three replicates for every tested sample concentration, control, and blank were measured. The DPPH radical scavenging ability was obtained using the following equation:(7)$$\mathrm{Scavenging}\;\mathrm{effect}\;\left(\%\right)=\left[1-\frac{A_{\mathrm{sample}\;517\;\mathrm{nm}}-A_{\mathrm{control}\;517\;\mathrm{nm}}}{A_{\mathrm{blank}\;517\;\mathrm{nm}}}\right]\times 100$$
where A_sample 517nm_ is the absorbance of the samples, A_control 517nm_ is the absorbance of the control (NADH was substituted with distilled water), and A_blank 517nm_ is the absorbance of the blank (samples were substituted with distilled water).

### 2.8. Cytotoxicity Assay

The cytotoxicities of chitosan, chitosan derivatives, and chitosan nanoparticles on four kinds of cells (three kinds of cancer cells, including MCF-7, BGC-823, and HEPG-2, and normal cells HTR-8) were conducted via CCK-8 assay using the previous method in vitro [3]. Taking MCF-7 as an example, MCF-7 cells were seeded on 96-well flat-bottom culture plates with the concentration of 1.0 × 10^5^ cells in 100 µL of RPMI medium, including the mixture of penicillin and streptomycin (1%) and fetal calf serum (10%), and incubated at 37 °C under a CO_2_ atmosphere (5%). Twenty-four hours later, every tested sample with a series of concentrations were added to cells, respectively. After incubation for 24 h, 10 µL of a CCK-8 solution was dropped into each well and the plates were incubated for a further 24 h. The absorbance of every well was recorded using a microplate reader (DNM-9602G, Thermo Multiskan Ascent, USA) at 450 nm. The cytotoxicities were calculated using the following formula:(8)$$\mathrm{Inhibition}\;\mathrm{rate}\;\left(\%\right)=\left(1-\frac{{A\;}_{\mathrm{sample}}-A_{\;\mathrm{blank}}}{A_{\mathrm{negative}}-A_{\mathrm{blank}}}\right)\times 100$$
where A_sample_ is the absorbance of samples (containing cells, CCK-8 solution, and sample solution), A_blank_ is the absorbance of blank (containing RPMI medium and CCK-8 solution), and A_negative_ is the absorbance of negative (containing cells and CCK-8 solution).

### 2.9. Statistical Analysis

Every experiment was repeated three times, and the data were reported as mean ± standard deviation (SD), *n* = 3. Significant difference analysis was determined according to Scheffe’s multiple range test. Significant differences (*p* < 0.05) between the means were determined using Duncan’s multiple range tests.

## 3. Results and Discussions

### 3.1. Characterization of Nanoparticles

#### 3.1.1. FTIR Spectra

Figure 1 shows the spectra of chitosan, chitosan derivatives, and chitosan nanoparticles, respectively. For chitosan, the main infrared absorption peaks appear at 3416 cm^−^^1^ (the O-H and N-H stretching vibrations), 2876 cm^−1^ (the -CH symmetric stretching vibration), 1655 cm^−1^ (the amino group bending vibration), and 1078 cm^−1^ (the C-O stretching vibration of the glucosamine ring) [16]. Compared to chitosan, the spectrum of HACC illustrates the successful introduction of the quaternary ammonium salt group. In particular, the characteristic peak of HACC located at 1480 cm^−1^ represents the stretching vibration of the trimethylammonium group [26]. At the same time, the characteristic peak located at 1626 cm^−1^ weakens appreciably due to a part of the primary amine being converted to the secondary amine. For HACF obtained after ion exchange, successful folic acid incorporation is confirmed by the new characteristic FTIR absorption bands at around 1716 cm^−^^1^ (the C=O stretching vibration of carboxyl in the folate molecule), 1550 cm^−^^1^, 1507 cm^−1^ (the C=C stretching vibration of the benzene ring in the folate molecule), and 768 cm^−1^ (the C-H stretching vibration of folate) [31]. As for HACNP, the interaction between TPP and trimethylammonium groups of HACC is proved by the new peaks at about 1565 cm^−1^ and 1375 cm^−1^, which is related to the formation of ammonium phosphate. In case of HACFNP, in addition to the singles located at 1487 cm^−1^ and 1416 cm^−^^1^ of the trimethylammonium group, a new peak belonging to the C=O stretching vibration of the carboxyl in folate molecule at 1692 cm^−1^ also appears. Compared to the blank nanoparticles (HACFNP), nanoparticles (HACAFNP) loading ADM based on HACF demonstrated a broadening peak at about 3421 cm^−1^, the characteristic peak appears at 1023 cm^−^^1^ (the C=O stretching vibration of ADM), and a new peak at 613 cm^−1^ (the C-H stretching vibration of ADM), indicating the ADM was loaded on the nanoparticles [3]. Therefore, the data confirmed the successful preparation of chitosan derivatives and chitosan nanoparticles.

#### 3.1.2. H NMR Spectra and DS of Chitosan Derivatives

Figure 2 shows the ^1^H NMR spectra of chitosan and chitosan derivatives. From the figure, the chemical shifts of pure chitosan are shown as follows: peaks at 3.12 ppm, 3.62–4.10 ppm, and 4.52 ppm are attributed to the hydrogen protons of [H2], [H3]–[H6], and [H1], respectively. For HACC, after quaternarization, the ^1^H NMR spectra of HACC appears with a new chemical signal at 3.22 ppm, which can be assigned to the hydrogen protons of the N^+^(CH_3_)_3_ group. Meanwhile, new peaks are easy to observe in the HACC spectra: 4.31 ppm (b), 2.78 ppm (a), and 2.54 ppm (c), which further prove the successful modification of chitosan via quaternarization. For the spectra of HACF, the peaks at 7.40 ppm (k,l) and 6.34 ppm (i,j) can be assigned to protons of the benzene ring of folate anion. Furthermore, the peak at 3.08 ppm representing the hydrogen protons of the N^+^(CH_3_)_3_ group can also be observed in the spectra of HACF. Therefore, the data sufficiently illustrates the successful preparation of chitosan derivatives.

The DS of chitosan derivatives were measured in accordance with the ^1^H NMR spectra of chitosan and chitosan derivatives. Taking HACF as an example, the DS of HACF was calculated by the ratio of integral of the hydrogen atom bonded to the N^+^(CH_3_)_3_ group and hydrogen atoms of the benzene ring. The specific measurement method was illustrated in the previous part. The DS of HACC and HACF was determined as 70.17% and 48.16%, respectively (Table 1).

#### 3.1.3. Hydrodynamic Diameter (nm), Zeta Potential (mV), and Nanoparticle Stability

The hydrodynamic diameter, polydispersity index (PDI), and zeta potential of different nanoparticles are significant indicators [32]. Therefore, in this paper, we evaluated the optimal preparation conditions based on hydrodynamic diameter, polydispersity index, and zeta potential. Firstly, particle size is an important factor affecting nanoparticles, and it is reported that the nanoparticles with particle sizes of 50–500 nm will be taken by endocytosis, while particles larger than 500 nm will be taken by phagocytosis [33]. In this study, the particle size of chitosan nanoparticles is shown in Figure 3a. As shown in the figure, the hydrodynamic diameters of HACNP, HACFNP, HACANP, and HACAFNP were 56.7 ± 0.74, 62.6 ± 2.15, 79.4 ± 0.60, and 85.6 ± 2.04, respectively. In particular, HACNP and HACFNP particles possess a relatively small hydrodynamic diameter and displayed hydrodynamic diameters of 56.7 ± 0.74, 62.6 ± 2.15, respectively. However, after the ADM loading on the nanoparticles, the hydrodynamic diameters of HACANP and HACAFNP increased significantly. In spite of this, the size of nanoparticles was much smaller than 500 nm, indicating that particles could be endocytosed by cells. Zeta potential is an essential indicator for the stability behavior of colloid, and it is measured by the electrostatic repulsion between adjacent particles with similar charge. It has been reported that nanoparticles with higher zeta potential (absolute value) are more stable due to the ability to resist aggregation [3,27]. The PDI and zeta potential of different chitosan nanoparticles are shown in Table 2. As shown in the table, the zeta potential of HACAFNP is +21.06 ± 0.96 mV, while its PDI is 0.23 ± 0.14, indicating that the nanoparticles are relatively stable and nanoparticle size distributions is relatively uniform. Moreover, it has been reported that positively charged chitosan nanoparticles are absorbed by the cells more easily because of the negative charge on the cell membrane [3]. Meanwhile, from Figure 3b, with the passage of time, the change of particle size of different nanoparticles was relatively small, which indicates that the nanoparticles had good stability and could serve as practical drug delivery system.

#### 3.1.4. Morphology Analysis

The morphology of nanoparticles was characterized via SEM (Figure 4). From the figure, we can see the surface morphology of the novel nanoparticles is spherical or almost spherical, and the novel nanoparticles exhibit a relatively small particle size. At the same time, the size of nanoparticles is corresponded with the particle size measured by a nanometer particle size measuring instrument.

### 3.2. Entrapped Efficiency and Drug Loading Analysis

The encapsulation efficiency and drug loading of freeze-dried nanoparticles (HACAFNP) loading ADM based on HACF were quantified under the optimal conditions. HACANP and HACAFNP possessed encapsulation efficiencies of 60.50% and 62.75%, respectively. Drug loading of HACANP and HACAFNP were 27.91% and 25.07%, respectively.

### 3.3. Swelling Degree Analysis

It has been reported that swelling degree is essential for chitosan nanoparticles. This is because chitosan nanoparticles with good swelling degree will facilitate more or less the release of ADM from the hydrogel matrix. From Figure 5, for HACAFNP, during the first 2 h, the swelling degree was negative because a part of the samples dissolved when the samples swelled in the swelling agent. In the later 10 h, the swelling degree was positive, and the swelling degree increased quickly. In other words, the porosity of chitosan nanoparticles increased with passage of time. With the increase of porosity of chitosan nanoparticles, the ADM in the nanoparticle was diffused through the pores of the nanoparticles.

### 3.4. In Vitro Release of Chitosan Nanoparticles

The objective of prepared nanoparticles loading ADM was to obtain a drug delivery system that could ultimately increase the residence time of a drug. Thus, the release of nanoparticles was compared to the diffusion of free ADM in an aqueous solution (control) through a phosphate buffer. From Figure 6, for free ADM, during the first 4 h, the release was relatively quick, with the cumulative release percentage of 73.25%. In the later 44 h, the release was very slow, and the release almost ended at 24 h, with a maximum of 83.56%. However, the cumulative release percentages of HACANP and HACAFNP in the release medium increased slowly during the first 4 h, with cumulative release percentages of 18.46 and 19.86%, respectively. Considering the swelling degree of the nanoparticles, the results showed that the ADM in the nanoparticles was diffused through the pores of the nanoparticles. At the same time, the dissolution of a small number of nanoparticles can also lead to drug release. In the later 44 h, the cumulative release percentages of HACANP and HACAFNP still increased slowly. In particular, when the release times were 6, 12, 24, 36, and 48 h, the cumulative release percentages of HACAFNP were 25.15%, 26.47%, 29.76%, 36.87%, and 40.15%, respectively. In general, the profile of ADM release of chitosan nanoparticles presents two phases: an initial burst release period where more than 18% of the entrapped ADM was released during the first 4 h, and a slow release period in which about 40% of the ADM was released in a continuous way during the following 40 h. In the initial burst release period, on the one hand, the dissolution of a small number of nanoparticles resulted the quick release of the drug. On the other hand, when prepared nanoparticles dispersed in water, the hydrogen bonds could form in the most superficial regions of the nanoparticles, accelerating the free diffusion of the encapsulated ADM [20]. In the slow release period, it took a long time for drug content trapped in the inner parts of the nanoparticles to diffuse through the gel-like layer that forms on the surface [3,20]. Meanwhile, it has been reported that the mechanism of slow release is related to the rigid and hydrophobic core of chitosan nanoparticles [20,34]. The special structure caused the release of ADM, which was greatly restricted, so slow release behavior from the core was observed. Therefore, our observations could indicate that the novel nanoparticles were stable in the release media, which can control drug release and prolong drug efficacy.

### 3.5. Antioxidant Activity Analysis

Oxidative stress is a pathological state due to the imbalance between oxidants and antioxidants, which can result in many diseases such as neurodegenerative disease, cancer, cardiovascular disease, and so on [35]. Therefore, it is very important for drug carriers to have good antioxidant activity. In this study, two antioxidant activities, including superoxide radical scavenging activity and the DPPH radical scavenging ability of chitosan, chitosan derivatives, and chitosan nanoparticles, were evaluated in vitro. The superoxide radical scavenging activity of tested samples is shown in Figure 7a. From the figure, several conclusions can be found as follows: firstly, chitosan and HACC have poor superoxide radical scavenging activity, with scavenging effects of 15.19% and 23.50% when the concentration was 1.6 mg/mL. However, after the introduction of folic acid via ion exchange, the HACF showed an enhanced superoxide radical scavenging activity of 51.23% with a concentration of 1.6 mg/mL. For all samples to be measured, the sample concentrations were positively correlated with clearance of the superoxide radical scavenging activity. Taking CS as an example, the scavenging effects were 3.19, 3.59, 7.43, 12.85, and 15.19% when the sample concentrations were 0.1, 0.2, 0.4, 0.8, and 1.6 mg/mL, respectively. Finally, the decreased superoxide radical scavenging activities were found for HACAFNP, HACANP, HACFNP, HACF, ADM, HACC, and CS with values of 72.44, 69.50, 58.65, 51.23, 44.56, 23.50, and 15.19%, respectively. The DPPH radical scavenging activity of chitosan, chitosan derivatives and chitosan nanoparticles with different concentrations are shown in Figure 7b. From Figure 7b, a growing tendency is found for CS, ADM, HACC, HACNP, HACANP, HACF, HACFNP, and HACAFNP with DPPH radical scavenging abilities of 68.29, 63.54, 60.03, 50.12, 49.44, 30.15, 24.14, and 23.54% when the concentration was 1.6 mg/mL. It is critical for nanomaterials with good biological activity to avoid infection and immunoreaction [21,31]. Based on the above analyses from Figure 7, HACANP and HACAFNP with enhanced radical scavenging activity may inhibit the lipid peroxidation inside the cell and serve as a practical drug carrier.

### 3.6. Cytotoxicity Analysis

Figure 8 shows the inhibition index values of a series of concentrations with different treatments (CS, HACC, HACF, HACNP, HACFNP, HACANP, and HACAFNP) employed to three kinds of cancer cells. From Figure 8a, we find that CS, HACC and HACF have relatively weak inhibitory effects on MCF-7 cells. However, after the loading of ADM, the enhanced inhibitory abilities on MCF-7 cells of HACANP and HACAFNP were very significant. Specifically, the rules were as follows: firstly, the inhibitory effects of HACANP, HACAFNP, and free ADM to MCF-7 cells both illustrated a dose dependence manner. For instance, the inhibitory indices of HACAFNP were 40.89, 66.87, 79.66, 85.27, and 87.34% when the series of concentrations were 25, 50, 100, 200, and 400 μg/mL, respectively. Secondly, compared to free ADM, nanoparticles (HACAFNP) loading ADM based on HACF had more a significant inhibitory effect on MCF-7 cells. Specifically, compared to free ADM with the index 73.21% at the concentration of 400 μg/mL, the inhibitory index of HACAFNP was 87.35%. From Figure 8b, similarly, it was found that the inhibitory abilities on BGC-823 of all samples were positively correlated with the concentrations. Moreover, nanoparticles (HACAFNP) possessed the strongest inhibitory ability of 79.40% at the concentration of 400 μg/mL. The inhibitory effect on HEPG-2 cells was also measured and the result is presented in Figure 8c. From the figure, it can be seen that the inhibitory effect against HEPG-2 cells of HACAFNP had been obviously improved compared with free ADM. In addition, as previously described, the inhibitory effect increased with the augment of sample concentration. In a nutshell, several findings can be drawn from Figure 8. Firstly, chitosan nanoparticles loading ADM (HACANP, HACAFNP) possess significant antitumor activities at all tested concentrations, which can confirm that synergistic effects existed in doxorubicin and nanoparticles [34]. Secondly, the inhibitory abilities against cancer cells of all samples are positively correlated with the concentrations. Thirdly, the antitumor activities decreased in the order of HACAFNP > ADM > HACANP. Although the drug loading capacity of HACAFNP (25.07%) was lower than HACANP (27.91%), its inhibitory effect against MCF-7 cells with the concentration of 400 μg/mL (87.35%) was higher than HACANP (79.66%). It should be related to the bioactivity of folic acid. Folic acid is attractive as a ligand for targeting cancer cell membranes due to the over-expression of folic acid receptor on many human cancer cell surfaces [31]. In a nutshell, we can conclude that the severe decrease in cell viability is attributed mainly to ADM released from HACANP and HACAFNP. Compared to free ADM, HACAFNP exhibited higher antitumor activity. The same trend was observed in the studies where doxorubicin loaded on chitosan-protamine nanoparticles triggered apoptosis, which might be connected to the interaction between the chitosan that forms the nanoparticles and the cells [31,36]. The self-activity and persistent release of chitosan nanoparticles enhanced the antitumor activity.

Meanwhile, the cytotoxicity assay of tested samples on normal HTR-8 cells was investigated to explore the biocompatibility of nanomaterials. In general, compounds are considered nontoxic or low in toxicity when the cell inhibition rate is less than 20%. From Figure 9, the inhibitory effects on HTR-8 cells treated with CS were about 0–10% at all tested concentrations. Meanwhile, the inhibitory effect of blank nanomaterials without ADM was low and the inhibitory index was about 10–22%. However, the inhibitory effect of free ADM on normal HTR-8 cells was significantly higher, and the inhibitory index exceeded 57% when the tested concentration was 400 μg/mL. In other word, free ADM is more toxic to normal cells. Fortunately, after the loading of ADM on chitosan nanomaterials, there is a remarkable reduction in the inhibitory index at different tested sample concentrations. Taking HACAFNP as an example, when the tested concentration was 400 μg/mL, the inhibitory index was 29.37%. Therefore, chitosan nanomaterials could be considered as practical biomaterials with dramatically enhanced biological activity and reduced cytotoxicity for drug loading/release in a delivery system instead of traditional drug delivery systems.

## 4. Conclusions

In this paper, nanoparticles (HACAFNP) loading ADM based on HACF were synthesized via ionic gelation with the aid of TPP. The characterization of nanoparticles was confirmed by means of FTIR, DLS, ELS, and SEM. The surface morphology of novel nanoparticles was spherical or almost spherical, and the novel nanoparticles exhibited a relatively small hydrodynamic diameter (85.6 ± 2.04 nm), uniform distribution (PDI of 22.93 ± 1.37%) and positive zeta potential (+21.06 ± 0.96 mV). Due to the relatively stronger positive zeta potential, the change in particle size at different times was relatively small, which indicated that the nanoparticles had good stability. The entrapped efficiency and drug loading of HACAFNP were determined to be 62.75% and 25.07%, respectively. Meanwhile, the drug release of novel nanoparticles was assayed in vitro and the result indicated that the novel nanoparticles were stable in the release media, which can control drug release and prolong drug efficacy. Afterward, the antioxidant efficiencies of the novel nanoparticles were assayed. In particular, the target nanoparticles exhibited significant enhancement in superoxide radical scavenging activity and DPPH radical scavenging activity. The cytotoxicities against three kinds of cancer cells (MCF-7, BGC-823, and HEPG-2) were estimated in vitro, and results showed the target nanoparticles inhibited the growth of cancer cells. It’s worth noting that the inhibition index of HACAFNP against BGC-823 cells was 71.19% with the sample concentration of 25 μg/mL, which was much higher than the inhibitory effect of free ADM. In addition, the cytotoxicity against normal cells (HTR-8) was estimated in vitro and the results indicated that after the loading of ADM on nanomaterials there was a remarkable reduction in the inhibitory index at different tested sample concentrations. It was demonstrated that the novel nanoparticles with dramatically enhanced antioxidant and antitumor activities, reduced cytotoxicity, and steady drug release could be used as practical candidates for drug loading/release in a delivery system.

## Data Availability

All data are contained in the manuscript.

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
