# Peer review of "Synthesis, Characterization, and Evaluation of Nanoparticles Loading Adriamycin Based on 2-Hydroxypropyltrimethyl Ammonium Chloride Chitosan Grafting Folic Acid"

_polymers, 2021, doi:10.3390/polym13142229_

Round 1

Reviewer 1 Report

The paper submitted by Mi et al. deals with the preparation and characterization of different drug-loaded folic acid-functionalized chitosan-based nanoparticles. The proposed strategy is not new but the obtained formulations are of interest as drug delivery systems for active targeting.

Even if there are some language errors, the manuscript is clear and the conclusions are supported by the results. The paper it will be publishable after some corrections:

  1. The full name of the products or samples must be provided in the abstract.
  2. The introduction section must be completed with several new articles based on functionalized CS-based nanoparticles: https://doi.org/10.1016/j.ijbiomac.2019.12.247; https://doi.org/10.1016/j.msec.2019.109828
  3. L121: “as previously reported”
  4. L128: “The precipitate was purified…”
  5. L198: delete “the with”
  6. An elemental analysis should be performed in order to calculate the functionalization degree with folate.
  7. Complete the caption of fig 2 with the pH value
  8. Before the in vitro release studies, the authors must carry out some swelling tests. Ionic crosslinking does not confer a high mechanical strength and therefore it is possible that the disintegration of the particles occurs.
  9. L409: I think that the difference of 2% between the loading capacity of HACANP and HACAFNP is not so big as the authors state in this sentence
  10. The inhibition index given in fig 6 and 7, is equivalent to the inhibition rate given in experimental section?! If yes, the authors must add the units (%). They need also to indicate which is the threshold, in terms of inhibition index, between cytotoxicity and non-cytotoxicity and then to reformulate the discussion of these results. It seems that an inhibition index higher than 20%, observed for the blank NPs, indicate a cytotoxic level and I don’t know why this cytotoxicity appears for NPs based only on a biocompatible polymer, as the chitosan.    
  11. The expression of the PDI values is quite unusual. Currently, PDI values are the between 0 and 1.
  12. The functionalization of the CS with folate is a clear advantage in order to obtain a targeted drug delivery but the authors have not performed cellular uptake studies in order to demonstrate this advantage with respect to the non-functionalized samples.

Author Response

Polymers

No.: polymers-1270645

Title: Synthesis, characterization, and evaluation of nanoparticles loading adriamycin based on 2-hydroxypropyltrimethyl ammonium chloride chitosan grafting folic acid

Dear editor,

Thank you for your letter and for the reviewers’ comments concerning our manuscript entitled “Synthesis, characterization, and evaluation of nanoparticles loading adriamycin based on 2-hydroxypropyltrimethyl ammonium chloride chitosan grafting folic acid”. Those comments are all valuable and very helpful for revising and improving our paper. We have studied comments carefully and have made corrections which we hope meet with approval. The main corrections in the manuscript and the responds to the reviewers’ comments are as following:

The paper submitted by Mi et al. deals with the preparation and characterization of different drug-loaded folic acid-functionalized chitosan-based nanoparticles. The proposed strategy is not new but the obtained formulations are of interest as drug delivery systems for active targeting. Even if there are some language errors, the manuscript is clear and the conclusions are supported by the results. The paper it will be publishable after some corrections:

1.The full name of the products or samples must be provided in the abstract.

Answer: Thank you for your kind suggestions and according to your recommendation, we have provided the full name of the products or samples. Very sorry for this mistake.

2.The introduction section must be completed with several new articles based on functionalized CS-based nanoparticles: https://doi.org/10.1016/j.ijbiomac.2019.12.247; https://doi.org/10.1016/j.msec.2019.109828.

Answer: Thank you for your kind suggestions and according to your recommendation, we have described the new articles based on functionalized CS-based nanoparticles (Lines 71-78). Thank you for your kind suggestions and we hope meet with approval.

  1. L121: “as previously reported”

Answer: Thank you for your kind suggestions and according to your recommendation, we have corrected the phrase (Line 124). Thank you for your kind suggestions and we hope meet with approval.

  1. L128: “The precipitate was purified…”

Answer: Thank you for your kind suggestions and according to your recommendation, we have corrected the phrase (Line 131). Thank you for your kind suggestions and we hope meet with approval.

  1. L198: delete “the with”

Answer: Thank you for your kind suggestions and according to your recommendation, we have deleted the word (Line 131). Very sorry for this mistake.

  1. An elemental analysis should be performed in order to calculate the functionalization degree with folate.

Answer: Thank you for your kind suggestions and according to your recommendation, we added the 1H NMR spectra of chitosan derivatives. The degrees of substitution of chitosan derivatives can be calculated by 1H NMR spectra and elemental analysis. In this paper, the DS of chitosan derivatives were measured in accordance with the 1H NMR spectra of chitosan and chitosan derivatives. Taking HACF as an example, the functionalization degree with folate was calculated by the ratio of integral of the hydrogen atom bonded to N+(CH3)3 group and hydrogen atoms of benzene ring. The specific measurement method was illustrated in the paper (Lines 308-328). Thank you for your kind suggestions and we hope meet with approval.

  1. Complete the caption of fig 2 with the pH value

Answer: Thank you for your kind suggestions and according to your recommendation, we have completed the caption of fig 2 with the pH value (Line 357-358). Very sorry for this mistake.

  1. Before the in vitro release studies, the authors must carry out some swelling tests. Ionic crosslinking does not confer a high mechanical strength and therefore it is possible that the disintegration of the particles occurs.

Answer: Thank you for your kind suggestions and according to your recommendation, we have carried out swelling tests (Line 209-217, 374-383). Thank you for your kind suggestions and we hope meet with approval.

  1. L409: I think that the difference of 2% between the loading capacity of HACANP and HACAFNP is not so big as the authors state in this sentence

Answer: Thank you for your kind suggestions and according to your recommendation, we have rewritten the sentence. Thank you for your kind suggestions and we hope meet with approval.

  1. The inhibition index given in fig 6 and 7, is equivalent to the inhibition rate given in experimental section?! If yes, the authors must add the units (%). They need also to indicate which is the threshold, in terms of inhibition index, between cytotoxicity and non-cytotoxicity and then to reformulate the discussion of these results. It seems that an inhibition index higher than 20%, observed for the blank NPs, indicate a cytotoxic level and I don’t know why this cytotoxicity appears for NPs based only on a biocompatible polymer, as the chitosan.

Answer: Thank you for your kind suggestions. The inhibition index given in fig 6 and 7, is equivalent to the inhibition rate given in experimental section, and we have added the units (%) and indicated the threshold value between cytotoxicity and non-cytotoxicity (Lines 489-497). The inhibition rate of chitosan nanoparticles at the highest concentration was measured after many tests, and the inhibition was about 20%, which could be considered as low toxicity. Thank you for your kind suggestions and we hope meet with approval.

  1. The expression of the PDI values is quite unusual. Currently, PDI values are the between 0 and 1.

Answer: Thank you for your kind suggestions and according to your recommendation, we have expressed the PDI between 0 and 1 (Lines 349-355), and the table were as follows. Thank you for your kind suggestions and we hope meet with approval.

Table 2. The polydispersity index (PDI) and zeta potential (mV) of chitosan. nanoparticles.

Samples

Hydrodynamic diameter (nm)

PDI

Zeta potential (mV)

HACNP

56.70.74

0.210.16

+24.670.34

HACFNP

62.62.15

0.220.06

+23.541.06

HACANP

79.40.60

0.240.09

+23.831.24

HACAFNP

85.62.04

0.230.14

+21.060.96

  1. The functionalization of the CS with folate is a clear advantage in order to obtain a targeted drug delivery but the authors have not performed cellular uptake studies in order to demonstrate this advantage with respect to the non-functionalized samples.

Answer: Thank you for your kind suggestions. In this paper, we mainly studied the antioxidant activity, slow-release effect and anti-tumor effect of doxorubicin-loaded nanoparticles based on 2-hydroxypropyltrimethyl ammonium chloride chitosan grafting folic acid. As for the targeting effect of folic acid, we will consider this point in the following experiments because the experimental conditions have not been studied. Thank you for your kind suggestions and we hope meet with approval.

The revised manuscript has been submitted to your journal. We hope that the responses and the revised manuscript adequately address your concerns and those of the reviewers and that this revised version is now acceptable for publication. Thank you for your time and concerns. If you have any questions, please feel free to contact us.

Yours sincerely

Zhanyong Guo

Yantai Institute of Coastal Zone Research, Chinese

Reviewer 2 Report

The manuscript is very interesting. However, I have few comments:

  1. What was the soruce of chitosan? Shrimp?
  2. FTIR - what is the resultion of equipment?
  3. SEM - please put more details about analysis (covered by gold? magnification, other parameters of analysis)
  4. 2.4 and 2.5 Coefficient should be written by capital R as R2
  5. Fig. 3 the legend is not visible (red color)
  6. Fig. 7 "3.3. Formatting of Mathematical Components." I dont understand.
  7. There is lack of statistical analysis (p<0.05). Please provides needed details.

Author Response

Polymers

No.: polymers-1270645

Title: Synthesis, characterization, and evaluation of nanoparticles loading adriamycin based on 2-hydroxypropyltrimethyl ammonium chloride chitosan grafting folic acid

Dear editor,

Thank you for your letter and for the reviewers’ comments concerning our manuscript entitled “Synthesis, characterization, and evaluation of nanoparticles loading adriamycin based on 2-hydroxypropyltrimethyl ammonium chloride chitosan grafting folic acid”. Those comments are all valuable and very helpful for revising and improving our paper. We have studied comments carefully and have made corrections which we hope meet with approval. The main corrections in the manuscript and the responds to the reviewers’ comments are as following:

The manuscript is very interesting. However, I have few comments:

  1. What was the soruce of chitosan? Shrimp?

Answer: Thank you for your kind suggestions. Chitosan with the average molecular weight (MW) of 200 kDa and the degree of deacetylation of 85% was obtained from Qingdao Baicheng Biochemical Corp. (China). And chitosan was obtained from commercial sources and used without further purification. And we have learned that the company extracted the chitosan by shrimp shells. Thank you for your kind suggestions and we hope meet with approval.

  1. FTIR - what is the resolution of equipment?

Answer: Thank you for your kind suggestions and according to your recommendation, we have expressed the resolution of equipment (Lines 170-172). Thank you for your kind suggestions and we hope meet with approval.

  1. SEM - please put more details about analysis (covered by gold? magnification, other parameters of analysis)

Answer: Thank you for your kind suggestions and according to your recommendation, we have put more details about analysis (Lines 192-195). Thank you for your kind suggestions and we hope meet with approval.

  1. 4 and 2.5 Coefficient should be written by capital R as R2

Answer: Thank you for your kind suggestions and according to your recommendation, we have written by capital R as R2 (Line 202, 226). Very sorry for this mistake.

  1. 3 the legend is not visible (red color)

Answer: Thank you for your kind suggestions and according to your recommendation, we changed the color of the legend to make it clearer (Figure 4. The SEM of chitosan nanoparticles). Very sorry for this mistake.

  1. 7 "3.3. Formatting of Mathematical Components." I dont understand.

Answer: Thank you for your kind suggestions and according to your recommendation, we have deleted the part of "3.3. Formatting of Mathematical Components.". Very sorry for this mistake. Thank you for your kind suggestions and we hope meet with approval.

  1. There is lack of statistical analysis (p<0.05). Please provides needed details.

Answer: Thank you for your kind suggestions and according to your recommendation, we have provided needed details of statistical analysis. Taking Fig. 8 as an example, letters (a-e) that are different in Fig. 8, according to various sample concentrations, differ statistically between different samples (P < 0.05). Thank you for your kind suggestions and we hope meet with approval.

The revised manuscript has been submitted to your journal. We hope that the responses and the revised manuscript adequately address your concerns and those of the reviewers and that this revised version is now acceptable for publication. Thank you for your time and concerns. If you have any questions, please feel free to contact us.

Yours sincerely

Zhanyong Guo

Yantai Institute of Coastal Zone Research, Chinese

Round 2

Reviewer 1 Report

The authors have answered to almost all my suggestions. However, they need to revise:

1. the reference 18 according to the following link: https://www.sciencedirect.com/science/article/pii/S014181301937151X?via%3Dihub

2. line 71: "Iurciuc-Tincu et al." instead of "Marcel et al."

3. line 74: "Rata et al." instead of "Anca et al."

Author Response

  1. the reference 18 according to the following link: https://www.sciencedirect.com/science/article/pii/S014181301937151X?via%3Dihub

Answer: Thank you for your kind suggestions and according to your recommendation, we have corrected the reference 18 (Lines 589-591). Very sorry for this mistake.

  1. line 71: "Iurciuc-Tincu et al." instead of "Marcel et al."

Answer: Thank you for your kind suggestions and according to your recommendation, we have corrected this part (Lines 71-74). Very sorry for this mistake. Thank you for your kind suggestions and we hope meet with approval.

  1. line 74: "Rata et al." instead of "Anca et al."

Answer: Thank you for your kind suggestions and according to your recommendation, we have corrected this part (Lines 74-77). Very sorry for this mistake. Thank you for your kind suggestions and we hope meet with approval.

Reviewer 2 Report

Thank you for the corrections. I accept the paper.

Author Response

Thank you for the corrections. I accept the paper.

Answer: Thank you for your approval.
